# Effect of Varying Inclusion Levels of Fossil Shell Flour on Growth Performance, Water Intake, Digestibility and N Retention in Dohne-Merino Wethers

**DOI:** 10.3390/ani9080565

**Published:** 2019-08-16

**Authors:** Olusegun O Ikusika, Conference T. Mpendulo, Titus J Zindove, Anthony I Okoh

**Affiliations:** 1Department of Livestock and Pasture Science, Faculty of Science and Agriculture, University of Fort Hare, Alice 5700, South Africa; 2SAMRC Microbial Water Quality Monitoring Centre, University of Fort Hare, Alice 5700, South Africa; 3Department of Animal Science, Faculty of Agriculture and Environmental Science, Bindura University of Science Education, Bindura 263, Zimbabwe; 4Applied and Environmental Microbiology Research Group (AEMREG), Department of Biochemistry and Microbiology, University of Fort Hare, Alice 5700, South Africa

**Keywords:** Fossil shell flour, growth performance, digestibility, nitrogen utilization, Dohne-Merino

## Abstract

**Simple Summary:**

With the recent negative public opinion on chemical-based feed additive in many nations of the world (for the health implication and environmentally hazard it posed), naturally occurring feed additive are urgently needed to replace and support the sustainable development in livestock production. The potential of Fossil shell flour as performance enhancer was investigated in this study. The major finding was that fossil shell flour increased growth performance, apparent nutrient digestibility, N retention and make minimum use of water in Dohne-Merino wethers. Hence it could be an alternative to chemical-based feed additive in livestock production.

**Abstract:**

This study was carried out to determine the effect of varying levels of Fossil shell flour (FSF) supplementation on growth performance, water intake, digestibility and N retention in Dohne Merino sheep pursuant to establishing the optimum inclusion rate of this supplement in Dohne Merino diets. Sixteen Dohne-Merino wethers (18 ± 1.5 kg body weight) were used in a complete randomized design with four animals per treatment. Sheep were fed a basal diet without FSF addition (control, T1), or with the addition of FSF (2%, T2), (4%, T3) or (6%, T4) of the diet for 105 days. Treatment 3 (4% FSF) has the highest values of dry matter intake, total weight gain, N retention and for most of the apparent digestibility nutrients (CP, EE and Ash) compared to treatment T1, T2 and T4(*p* < 0.05). The urinary and fecal N excretion also significantly decreased in the FSF treated diets compared to the control (*p* < 0.05). Water intake values were highest in control and were significantly (*p* < 0.05) different from those in treatments 2 and 4, but not to treatment 3. It is concluded that 4% inclusion rate of FSF will give the best improvement on growth performance, diet digestibility and N retention of Dohne-Merino sheep. Also, the addition of FSF in the diets of sheep is a safe natural additive that can help to reduce environmental pollution by reducing fecal and urinary N excretion.

## 1. Introduction

The use of growth promoters in farm animal diets is growing at an increasing rate [1]. Growth promoters are used to improve feed efficiency and the growth performance of farm animals. Rumen regulation through the use of growth promoters is one of the most important methods for improving feed efficiency and, thus, growth performance. Likewise, feed additives in ruminant’s nutrition has the potential to increase dry matter intake (DMI), feed conversion efficiency (FCE) and animal productivity [1,2]. There is a wide range of feed additives that include antibiotics, probiotics, antioxidants, enzymes, prebiotics, organic acids, mycotoxin binders, hormones, beta agonist, defaunation agents, essential oil and herbal feed additives, most of which are chemical based [2].

Although the use of growth promoters as feed additives has been a hallmark of modern animal husbandry, in recent years there have been increased concerns on chemical residues in meat and other animal products as a result of these chemical based feed additives [3,4]. There is also increase in ecological risk because of the accumulation of veterinary antibiotics residue in animal manure [5], bodies of water, sediments and soils [6,7]. Arikan [8] reported that antibiotics administered to farm animals either as growth promoter or medication were usually excreted without metabolism. Similarly, [9] observed that between 70 to 90% of tetracycline may be excreted as parent compounds through urine or feaces. 

Because of the possible risks of chemical-based growth promoters, there have been increased interest in natural growth promoters (NGPs). Several plants and plant extracts, enzymes, organic acids and oils have received considerable attention recently as possible NGPs that are eco-friendly [10]. One of the major drawbacks to the use of these NGPs is the time and cost involved in harvesting them. One NGP that could be useful as a cost effective, readily available, health and eco-friendly feed additive is Fossil shell flour. Fossil shell flour is a naturally occurring silicate rich substance with important physical and chemical characteristics that enable its uses recently as feed additive in livestock production. The substance is nontoxic, cheap, and readily available in large quantity in many countries [11]. The mineral constituent of dietary fossil shell flour as reported by [12,13,14] are as follows: Sodium, 923 mg/kg; Copper, 30 mg/kg; Zinc, 118 mg/kg; Iron, 7944 mg/kg; Magnesium, 69 mg/kg; Calcium, 0.22%; Magnesium, 0.11%; Potassium, 0.08%; Aluminum, 0.065%; Sulfate sulphate, 0.062%, Boron, 23 mg/kg, and Vanadium, 438 mg/kg.

In the study conducted by Emeruwa et al. [15] using West Africa dwarf sheep, it was observed that inclusion of Fossil shell flour in the diet statistically affected the average daily weight gain with the highest value (0.20 kg/day) observed for the sheep fed the 4% inclusion level of Fossil shell flour and the lowest value of (0.11 kg/day) for those on 6% inclusion of fossil shell flour. Likewise, Sarijit et al., [16] reported that addition of diatomaceous earth (another name for FSF) to animal feeds (3.2% inclusion) improve daily feed intake, weight gain and feed efficiency.

On the effects of FSF on digestibility, [15] Emeruwa, (2016) observed that crude protein (%) digestibility was significantly higher (82.79) for sheep on 4% inclusion level of fossil shell flour than other levels. This author also reported that although N intake and the fecal and urinary N excretion were not significantly different among the treatment groups, N retention (% N intake) was significantly higher (72.4%) for sheep on 2% inclusion level of Fossil shell flour than for the other groups.

There is paucity of information on feedlot performance of Dohne-Merino supplemented with varying levels of FSF, and no reports are available on the effect of FSF on the water intake of Dohne-Merino. Considering that the use of this feed additive is cost effective, health and eco-friendly, the objective of this study was to assess the effects of 4 levels of FSF inclusion on growth performance, water intake, digestibility and N retention of Dohne-Merino sheep. We hypothesized that the inclusion of FSF at varying levels could increase the growth performance, digestibility and N retention of Dohne-Merino sheep.

## 2. Materials and Methods

### 2.1. Ethical Approval

The handling and the use of the animals was approved by University of Fort Hare, Animal ethics and Use Committee [Approval number (MPE041IKU01)].

### 2.2. Study Site Description

The experiment was conducted at the small ruminant unit of the University of Fort Hare teaching and research farm (animal section), Alice, Eastern Cape, South Africa. The research farm is located at about Km 5 along Alice-Kings Williams town, which lies at longitude 26°50′ E and latitude of 32°46′ S. The annual rainfall is between 480–490 mm and temperature range between 24.6 °C and 11.1 °C (average is 17.8 °C) at the altitude of 535 m above sea level.

### 2.3. Animal, Experimental Design and Management

Sixteen Dohne-Merino wethers (6 months old) weighing 20 ± 1.5 kg on average were selected from a commercial farm in Mitford village Tarkastad, Eastern Cape of South Africa, and were used for this study in a completely randomized design. All the 16 wethers were raised at the same facility in the same area under the same environmental conditions (University of Fort Hare, Teaching and Research Farm, Animal Section, Alice 5700, RSA). The wethers were randomly allotted into four treatment (*n* = 4). They were individually housed (1.5 m × 1.5 m) in a well-ventilated roofed animal building with concrete floor. The pens have similar temperature, relative humidity and sunlight conditions. The experiment lasted for 105 days, excluding 14 days of adaptation period. The animals have access to sufficient clean and fresh water over the trial.

### 2.4. Experimental Diets

The diets for the animals consisted of concentrate and hay at 40: 60 ratio. The concentrate was made up of maize (8%), sunflower oil cake (10%), molasses (5%), wheat offal (15%), limestone (1.5%), salt 0.3% and sheep mineral-vitamin premix (0.2%), whereas the hay consisted of 30% teff and 30% Lucerne. All ingredients were thoroughly milled and mixed evenly together to form the basal diet. The feed was formulated to meet the nutritional (energy and protein) requirements of the used sheep [17]. The four dietary groups were: T1: Basal diet (Control); T2: Basal diet +2% FSF; T3: Basal diet +4% FSF, and T4: Basal diet +6% FSF. The animals were fed at 8:00 h and 15:00 h at 4% of the body weight (on dry matter (DM) basis). The Fossil shell flour (Food - Grade) was purchased from Eco-Earth (Pty) Ltd., Port Elizabeth, SA which produces this product under a license by Department of Agriculture, Forestry and Fisheries of South Africa.

### 2.5. Analytical Procedures

Dry matter content of the diets, orts and fecal samples was measured by drying samples in an air-forced oven at 135 °C for 24 h (method 930.15; [18]) Ash content was measured by placing samples into a muffle furnace at 550 °C for 5 h (method 938.08; [18]. Organic matter (OM) was measured as the difference between DM and the ash content. Nitrogen (N) was measured by the Kjeldahl method using Se as a catalyst and crude protein (CP) was calculated as 6.25 × N. Gross energy (GE) was measured using a bomb calorimeter (C200, IKA Works Inc., Staufen, Germany). Ether extracts (EE) were measured by weight loss of the DM on extraction with diethyl ether in Soxhlet extraction apparatus for 8 h (method 920.85) [18]. Crude fibre was determined by allowing the sample to boil with 1.25% dilute H_2_SO_4_, washed with water, further boiled with 1.25% dilute sodium hydroxide and the remaining residue after digestion was taken as crude fibre (method 978.10) as described by Thiex [19].

### 2.6. Feed Intakes and Growth Performance

During the 105 days of feeding trial, data on feed offered to each animal and the corresponding orts were recorded daily to estimate voluntary intake of DM and nutrients. Samples of feeds offered and orts were oven dried at 65 °C until a constant weight to determine DM concentration, and then ground to pass through 1-mm sieve (Wiley mill; Thomas Scientific, Philadelphia, PA, USA) and analyzed for organic matter (OM), CP, EE and CF by the procedures described above. The body weight (BW) of animals was individually recorded at the beginning of the trial, on weekly basis throughout the trial, and at the end of the experiment before the morning feeding. Feed intake, average daily gain (ADG) and feed efficiency were calculated from the data obtained.

### 2.7. Apparent Nutrients Digestibility and N Retention

Apparent digestibility coefficients of DM, OM, CP, EE and CF were determined by the total fecal collection method [20]. On day 91, 3 animals per treatment were placed into individual metabolism crates (0.5 m × 1.2 m), allowing feces and urine to be collected. The digestibility trial lasted for 14 days with 7 days for adaptation to metabolism crates and 7 days for the sample collection.

The amount of feed offered, refused, and feces were weighed daily and homogenized. A 10% sample of total feces was collected during a 7-day collection period as described by Ma et al. [21]. Urine was collected daily in buckets containing 100 mL of 10% (v/v) H_2_SO_4_. The volume was measured and a sample (10% of total volume) was collected and stored at −20 °C until analysis. Samples of feed, orts, feces, and urine were pooled to form a composite sample for each wether. Urinary N was analyzed by the Kjeldahl method [18], and N retention was calculated as daily N excretion (urinary N plus fecal N) subtracted from daily N intake. 

### 2.8. Statistical Analysis

The data on apparent digestibility and N retention were analyzed using the PROC MIXED of SAS (version 9.1; SAS Inst. Inc., Cary, NC, USA). Because the experimental design was completely randomized, the model included only the fixed effect of the diet (treatment). Repeated measures were used to analyze data on feed and water intake and growth parameters. The effects of diet (treatment), weeks, and their interactions were considered fixed, whereas the wether was considered random. Data are presented as mean ± standard error of the mean, and significant differences were accepted if *p* < 0.05, Orthogonal polynomial contrasts were used to test the linear and quadratic effects of the diet on the parameters measured.

## 3. Results and Discussion

### 3.1. Chemical Composition of the Experimental Diets

The chemical composition of the basal experimental diets is shown in Table 1.

### 3.2. Feed and Water Intake

As shown in Table 2, the inclusion of FSF in the diet of Dohne-Merino wethers had a significant effect (*p* < 0.05) on feed intake. Animals in Treatments 3 and 4 consumed more feed than animals in the control group (linear *p* = 0.04; quadratic *p* = 0.02) while animals in treatment 2, consumed the least feed. This shows that the addition of FSF to the diets of Dohne-Merino wethers at 4% inclusion rate will increase the daily feed intake. This is similar to the results obtained by Emeruwa et al. [15], who recorded the lowest feed intake in the group fed 2% of FSF (900 ± 9.0 g/d) and the highest value for the 6% inclusion group (1009 ± 9.0 g/d). Also, Mclean et al. [22] reported significant variations in the feed intake of calves fed varying levels of FSF. Sheep requires about 25 to 40 ounces per head per day of NaCl, and will eat less feed when deprived of NaCl [23,24]. Similarly, it has been observed that minerals such as Na, Ca, K and Mg increase the palatability of the diet and the feed intake of the animal [25,26]. This could be the reason why animals fed FSF-supplemented diets had greater feed intakes compared with the controls, because FSF had been reported to be rich in minerals, including Na (923 ppm), Ca (0.22%), Mg (0.11%) and K (0.11%) [12,13,14]. Feed efficiency was highest in T3 while the control treatment has the lowest value (*p* < 0.05). However, the control was not statistically different from T2 and T4. The greater feed intake in T3 group was especially notable from week 7 until the end of the trial (Figure 1). The absence of quadratic effect observed for various inclusion levels of FSF on nutrients intake shows that adding increasing levels of FSF up to 6% inclusion levels of the diets DM had no adverse effects on their consumption by the animals.

The water intake decreased linearly (*p* < 0.001) as the inclusion levels of FSF increased in the diets, except for 4% inclusion level (Table 2). The control group has the greatest water intake, while T4 has the lowest value. Although the control group consumed slightly more water than T3 group, differences were not significant (*p* > 0.05). Increased water intakes usuallly encourage sheep to eat more feed [27,28]. 

### 3.3. Growth Performance

The effects of varying inclusion levels of FSF on growth performance are shown in Table 2. Average daily weight gain was increased (linear *p* = 0.00; quadratic *p* = 0.01) and final weight was greater (linear *p* = 0.04) with increasing inclusion levels of FSF. The ADG was lowest for control group and greatest for T3. The lowest ADG and final BW in the control group (T1) is in agreement with the lower DMI recorded in the trial for this group. 

### 3.4. Nitrogen Utilization and Nutrients Apparent Digestibility.

Table 3 shows the nitrogen utilization and diet apparent digestibility in Dohne-Merino rams fed varying levels of FSF. Nitrogen intake, urinary N and N balance were not affected by the levels of FSF inclusion. However, fecal N was decreased (linear *p* = 0.03; quadratic, *p* = 0.03) as inclusion levels of FSF increased. This indicates that FSF could be used to reduce environmental pollution through reduction in fecal N excretion. This result was similar to the report obtained by Rajabi et al. [29] using pomegranate peel extract in fattening lambs as feed supplement. Vallejo et al. [30] reported that addition of natural feed additive such as xylanase (at 3 µL/g) to the diets of sheep could reduce excretion of N through fecal and urine. 

The linear positive effect of the treatment on daily retained N in the present study was similar to that reported by Soroor et al. [31] when fattening Mehraban lambs were fed *Echium amoenum* extract up to 1.5 mL/kg of diet DM. Conversely, this result was not in agreement with the result observed by Emeruwa et al. [15] using FSF as supplement in West African dwarf sheep, as they observed that N retention was greater in control compared to the treatments. However, both in this study and in the study reported by Adebiyi et al. [12], FSF has no significant effect on N balance and N retention. The differences among studies could be due to different experimental conditions. This study was conducted in a semi-arid region while Emeruwa et al. conducted their trial in a tropical region. Other factors, as dietary and breed differences could have also influenced the results. 

Increasing the inclusion levels of FSF had no effects on DM, OM, EE, and CF digestibility (Table 3), but it has a quadratic effect on apparent CP digestibility (*p* = 0.01). The diet supplemented with 4% FSF (T3) showed the greatest value for all nutrient’s digestibility and was significantly different (*p* < 0.05) from the control (T1) for all nutrients. However, there were no differences (*p* > 0.05) among T2, T3 and T4 treatments for CP and EE digestibility (Table 3). These results indicate that FSF enhanced the absorption of amino acids by the animal and thereby making them available at tissue levels [32]. These results are contrary to those obtained by Emeruwa et al. [15], who found that West African Dwarf sheep fed diets containing FSF at the same levels of inclusion than in the present study had numerically greater CP and EE digestibility values for control compared to other treatments, although differences did not reach the statistical significance. This difference in the results could be due to differences in breeds and diets composition. The lack of influence of FSF on DM, OM, EE and CF, may be attributed to similar digestibility of the experimental diets, particularly crude fibre digestibility [33]. Additionally, the relative similarity in the physical characteristics of the experimental diets may be the reason(s) for lack of no intake differences among the experimental sheep, because physical and nutrients composition of diets affect the feed intake [34]. 

## 4. Conclusions

The results from this study showed that adding 4% FSF to the diets of Dohne-Merino wethers gives a better improvement than other levels of FSF inclusion on DMI, feed efficiency, ADG, total weight gain, N retention and apparent nutrient digestibility of most of the nutrients. In addition, the wethers on this treatment consumed the lowest volume of water. These results suggested that inclusion of FSF at 4% in the diet could be potentially used as a performance enhancer.

## Figures and Tables

**Figure 1 animals-09-00565-f001:**
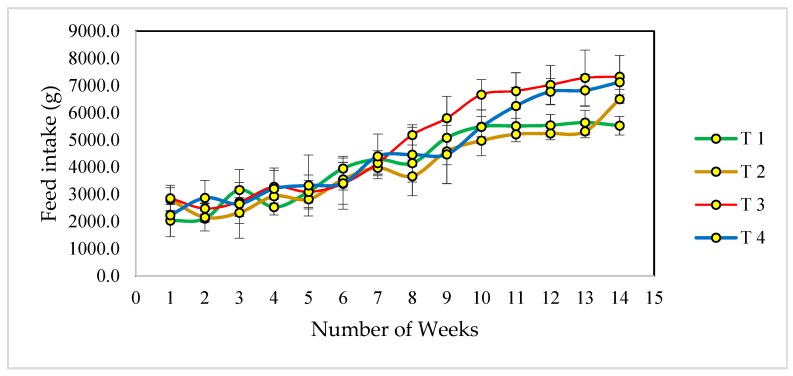
Graphical representation of feed intake of Dohne-Merino wethers fed varying levels of Fossil shell flour (FSF) shown as means ± standard errors. T1: 0% FSF diet; T 2: 2% FSF diet; T3: 4% FSF diet; T4: 6% FSF diet.

**Table 1 animals-09-00565-t001:** Ingredients and chemical composition of the basal diet fed to Dohne-Merino sheep.

Items	Percentage (%)
Maize	8
Sunflower oil cake	10
Molasses	5
Wheat bran	15
Limestone	1.6
Sheep mineral-vitamin premix	0.20
Salt	0.30
Grinded Lucerne hay	30
Grinded teff hay	30
Chemical composition	
Dry Matter (% as fed)	95.5
Organic Matter (%DM)	85.2
Energy (MJ/kg DM)	24.7
Crude Protein (% DM)	14.6
Ash (%DM)	10.3
Ether Extract	1.70
Crude Fibre	22.60

**Table 2 animals-09-00565-t002:** Effects of varying levels of FSF on growth performance and water intake on Dohne-Merino wethers.

Parameter	Level of FSF in Diet (% of DM)				SEM	*P*-Values	
	0	2	4	6		Linear	Quadratic
Feed Intake(g)	593	572	694	648	39.9	0.045	0.024
Feed Efficiency (g/g)	0.14	0.18	0.19	0.17	0.01	0.011	0.023
ADG (g/d)	84.7	92.9	121.4	105.4	9.53	0.000	0.011
Water Intake (L)	1.95	1.67	1.8	1.45	0.10	0.000	0.113
Initial Weight (Kg)	19.2	19.4	19.5	19.4	0.43	0.667	0.221
Final Weight (Kg)	28.0	28.6	31.3	29.7	0.48	0.043	0.322
Total Weight Gain (Kg)	8.32	9.20	12.0	10.3	0.38	0.042	0.232

**Table 3 animals-09-00565-t003:** Nitrogen utilization (g/d) and apparent digestibility (%) of Dohne Merino fed varying levels of Fossil shell flour.

Parameter	Level of FSF in Diet,% of DM				SEM	*P*-Values	
	0	2	4	6		Linear	Quadratic
**N Retention(g/d)**							
**N Intake**	9.83	11.07	13.96	10.24	0.44	0.433	0.954
**Fecal N**	1.94	1.31	1.23	1.48	0.47	0.033	0.033
**Urinary N**	0.79	0.32	0.41	0.58	0.07	0.856	0.842
**N Balance**	7.15	9.41	12.35	8.18	0.46	0.454	0.965
**Apparent Digestibility (%)**							
**Dry Matter**	64.50	63.83	72.92	62.83	2.74	0.375	0.311
**Organic Matter**	65.58	66.24	73.15	63.13	2.81	0.353	0.367
**Crude Protein**	70.62	81.83	85.24	77.16	1.65	0.192	0.011
**Ether Extract**	85.54	89.69	93.01	90.19	1.41	0.247	0.222
**Crude Fibre**	51.10	51.17	64.79	52.73	3.88	0.712	0.312

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
