# Peer review of "Effect of Varying Inclusion Levels of Fossil Shell Flour on Growth Performance, Water Intake, Digestibility and N Retention in Dohne-Merino Wethers"

_animals, 2019, doi:10.3390/ani9080565_

Round 1

Reviewer 1 Report

Thank you for addressing my concerns with the manuscript.

Reviewer 2 Report

The subject is certainly original and interesting, the manuscript is very clearly described. The results obtained sound very interesting and of sure impact for breeders.

This manuscript is a resubmission of an earlier submission. The following is a list of the peer review reports and author responses from that submission.

Round 1

Reviewer 1 Report

Minor points:

line 27 (abstract) please add T1 after 0%

line 60 (introduction) please add in between interest and natural

line 150 (materials and methods) and Feed efficiency correct and feed efficiency

second line, paragraph 3.4 "is significantly difference" please correct with "is significantly different

Discussion, paragraph 4.3 please delete the acronyms T1 and T3, when present, because it is also witten in full (treatment 1 and treatment 3). line 26 please delete the parenthesis after "treatments"

Paragraph 4.4, please delete [29] before Mpendulo

Paragraph 4.5 line 40 please correct "is significantly difference" with "is significantly different"

Please check that the numbering of the lines is progressive

Author Response

 Reviewer comment:line 27 (abstract) please add T1 after 0%.

Author's reply: I have added  'T' after 0%

Reviewer comment: Line 60 (introduction) please add in between interest and natural

Author's reply: I have added in between interest and natural.

Reviewer comment: line 150 (materials and methods) and Feed efficiency correct and feed efficiency.

Author's reply: Feed efficiency change to feed efficiency

Reviewer comment: second line, paragraph 3.4 "is significantly difference" please correct with "is significantly different.

Author's reply: I cannot find the phrase " is significantly difference" in paragraph 3.4 second line.

Reviewer comment: Discussion, paragraph 4.3 please delete the acronyms T1 and T3, when present, because it is also written in full (treatment 1 and treatment 3). line 26 please delete the parenthesis after "treatments

Author's reply: I have deleted the acronyms T1 and T3. in paragraph 4.3. I have also changed "T2" to "treatment 2". For uniformity with others. There is no parenthesis inline 26.

Reviewer comment: Paragraph 4.4, please delete [29] before Mpendulo

Author's reply: No [29] before Mpendulo in paragraph 4.4. Maybe it has been corrected.

Reviewer comment: Paragraph 4.5 line 40 please correct "is significantly difference" with "is significantly different"

Author's reply: I can't find paragraph 4.5 in this manuscript. However, I have changed all the "significantly difference" to "significantly different"

Reviewer comment: Please check that the numbering of the lines is progressive

Author's reply: numbering has been done progressively

Reviewer 2 Report

This manuscript needs a lot of revising.  There are many errors throughout.  I will start with tables and figures.

You mention table 2 having nutrients listed this is in Table 1

Why is water discussed separately data should be discussed as it appears in order in text.

kg not Kg

Figure 1 is impossible to differentiate treatment differences use a different symbol for each treatment

Table 3 is confusing.  What do you mean "urine intake and fecal intake. N retention is commonly measured in g.  Are the values for Nret %?  please correct If so this is N efficiency 

Figure 2 Y axis needs to line up again use symbols to define treatments.  

Figurre 3 needs to be redone based on previous comments regarding figures.  change the y axis to begin at 10 kg

All figures need to have a value for standard error

Feed intake ? Is the DM intake?

Throughout the manuscript- do not repeat the data presented in the tables. 

Numerical differences are not different and should not be discussed.

Do not present the p values if you state that the effects were similar.

Your statements regarding that goats that consumed the most DM drink the least amount of water does not make sense to me.  Water increases rate of passage and hence gut emptying and ultimately increases DMI Can this be correct?

Line 247 -253 is not correct.  While it is true that the control consumed the most water, it appears to be similar to the 4% treatment.  Can you conduct some mean separation to figure this out? Treatment 4 had the greatest feed intake, but also a ranked numerically similar to the control in water intake, but this needs to be checked.

Author Response

1. Reviewer comment: You mention table 2 having nutrients listed this is in Table 1

Author's reply: correction had been done. "Table 1" has been inserted and "Table 2" deleted.

2. Reviewer comment: Why is water discussed separately data should be discussed as it appears in order in text.

kg not Kg

Author's reply: It has been rearranged according to the order in the text. Water is discussed separately because it is not part of growth parameters. Also "Kg" has been changed to "kg"

3.Reviewer comment: Figure 1 is impossible to differentiate treatment differences use a different symbol for each treatment

Author's reply: Thank you for your observation, we don't want to agree with this observation. The different treatment is clearly differentiated in this figure. This difference in the treatment is shown by the different colors in the plot in this figure. Using different symbols will not make much difference.

4.Reviewer comment:Table 3 is confusing.  What do you mean "urine intake and fecal intake? N retention is commonly measured in g.  Are the values for Nret %?  please correct If so, this is N efficiency.

Author's reply: Table 3, has been arranged very well. Urine intake changed to "urinary N". fecal intake changed to "fecal N". N retention is measured in g/d. It is stated there.

5. Reviewer comment:Figure 2 Y axis needs to line up again use symbols to define treatments. 

Author's reply: Again we sincerely appreciate this observation. However, the different treatment is also clearly differentiated in this figure by the use of different colors in the plot and labeled accordingly in the legend. We do not understand what the reviewer meant by Y-axis line up, as a result, we couldn't act nor comment on this.

6. Reviewer comment:Figurre 3 needs to be redone based on previous comments regarding figures.  change the y axis to begin at 10 kg

Author's reply: Figure 3 has been redone, standard error added and y-axis began at 10kg

7.Reviewer comment: All figures need to have a value for standard error.

Author's reply: Values for standard error has been added to all figures.

8. Reviewer comment: Feed intake? Is the DM intake?

Author reply: The comment is not explicit enough. However, figure I showed the feed intake over 14 weeks while figure 2 showed the dry matter intake for the same period. thanks

9. Reviewer comment: Throughout the manuscript- do not repeat the data presented in the tables.

Author's reply: Data presented in the tables have been removed from the manuscript. That is no more repetition.

10.Reviewer comment: Numerical differences are not different and should not be discussed.

Author's reply: All numerical differences that were included in the discussion have been removed and sentences rephrased

11.  Reviewer comment: Do not present the p values if you state that the effects were similar.

Author's reply:p values deleted where it is been stated that effects were similar.

12.Reviewer comment: Your statements regarding that goats that consumed the most DM drink the least amount of water does not make sense to me.  Water increases rate of passage and hence gut emptying and ultimately increases DMI Can this be correct?

Author's reply: The statement has been puy straight and supported by relevant references.

 13. Reviewer comment: Line 247 -253 is not correct.  While it is true that the control consumed the most water, it appears to be similar to the 4% treatment.  Can you conduct some mean separation to figure this out? Treatment 4 had the greatest feed intake, but also a ranked numerically similar to the control in water intake, but this needs to be checked.

Author's reply: The correction has been made. More journal cited. Line 224-229. The statement has been put straight

Round 2

Reviewer 2 Report

This is an interesting manuscript. Many of my suggestions have been addressed, but not all.  English needs to be improved.  

Use greater for higher and lesser for lower.

You said you removed the p values that were not sig. However, there are many in line207-214. Please correct.  I would submit this for review by a native English speaker- it will improve the quality.  

Author Response

Reviewer comment:This is an interesting manuscript. Many of my suggestions have been addressed, but not all.  English needs to be improved. 

Author's reply: An English editor has gone through the manuscript and has made corrections.

Reviewer comment: Use greater for higher and lesser for lower.

Author's reply: greater has been used instead of higher and lesser instead of lower (Line 173).

Reviewer comment:You said you removed the p values that were not sig. However, there are many in line207-214. Please correct.  I would submit this for review by a native English speaker- it will improve the quality.  

Author's reply: The p values in  line 207-214 has all been removed.